# Hepatic Lipoprotein Metabolism: Current and Future In Vitro Cell-Based Systems

**DOI:** 10.3390/biom15070956

**Published:** 2025-07-02

**Authors:** Izabella Kiss, Nicole Neuwert, Raimund Oberle, Markus Hengstschläger, Selma Osmanagic-Myers, Herbert Stangl

**Affiliations:** 1Institute of Medical Chemistry, Center for Pathobiochemistry and Genetics, Medical University Vienna, Vienna 1090, Austria; izabella.kiss@meduniwien.ac.at (I.K.); nici.neuwert@gmx.de (N.N.); raimund.oberle@meduniwien.ac.at (R.O.); 2Institute of Medical Genetics, Center for Pathobiochemistry and Genetics, Medical University Vienna, Vienna 1090, Austria; markus.hengstschlaeger@meduniwien.ac.at

**Keywords:** lipoproteins, human, liver, organoids, iPSCs/ESCs, tissue culture, metabolism

## Abstract

Changes in hepatic lipoprotein metabolism are responsible for the majority of metabolic dysfunction-associated disorders, including familial hypercholesterolemia (FH), metabolic syndrome (MetS), metabolic dysfunction-associated fatty liver disease (MAFLD), and age-related diseases such as atherosclerosis, a major health burden in modern society. This review aims to advance the understanding of state-of-the-art mechanistic concepts in lipoprotein metabolism, with a particular focus on lipoprotein uptake and secretion and their dysregulation in disease, and to provide a comprehensive overview of experimental models used to study these processes. Human lipoprotein research faces several challenges. First, significant differences in lipoprotein metabolism between humans and other species hinder the reliability of non-human model systems. Additionally, ethical constraints often limit studies on human lipoprotein metabolism using tracers. Lastly, while 2D hepatocyte cell culture systems are widely used, they are commonly of cancerous origins, limiting their physiological relevance and necessitating the use of more physiologically representative models. In this review, we will elaborate on key findings in lipoprotein metabolism, as well as limitations and challenges of currently available study tools, highlighting mechanistic insights throughout discussion of these models. These include human tracer studies, animal studies, 2D tissue culture-based systems derived from cancerous tissue as well as from induced pluripotent stem cells (iPSCs)/embryonic stem cells (ESCs). Finally, we will discuss precision-cut liver slices, liver-on-a-chip models, and, particularly, improved 3D models: (i) spheroids generated from either hepatoma cancer cell lines or primary human hepatocytes and (ii) organoids generated from liver tissues or iPSCs/ESCs. In the last section, we will explore future perspectives on liver-in-a-dish models in studying mechanisms of liver diseases, treatment options, and their applicability in precision medicine approaches. By comparing traditional and advanced models, this review will highlight the future directions of lipoprotein metabolism research, with a focus on the growing potential of 3D liver organoid models.

## 1. Introduction—Current Concepts in Hepatic Lipoprotein Metabolism

Lipoprotein metabolism plays a central role in maintaining systemic lipid homeostasis, ensuring the efficient transport of lipids, lipid redistribution to different organs, and clearance. The liver serves thereby as a key regulatory hub, orchestrating lipoprotein synthesis and clearance to maintain the lipoprotein balance. After a meal, this process is initiated in the intestine, where chylomicrons, triglyceride-rich lipoproteins, are synthesized and where they enter the circulation. Subsequently they are processed by lipoprotein lipase (LPL), supplying peripheral organs with fatty acids. The resulting chylomicron remnant particles are cleared by hepatic receptors, such as low-density lipoprotein receptor (LDLR) and low-density lipoprotein -receptor related protein 1 (LRP1), contributing their lipid load to the synthesis of very low-density lipoprotein (VLDL) particles as part of the exogenous lipoprotein pathway [1,2]. In a similar way, triglycerides from VLDL particles are utilized by peripheral tissues, while their cholesterol-rich remnants/intermediate density lipoproteins (IDLs) are cleared by the liver as part of the endogenous lipoprotein pathway [3] (see also the review on VLDL particles in [4]). Remnants/IDLs are also further converted by action of LPL, hepatic lipase (HL), or cholesteryl ester transfer protein (CETP) to low-density lipoprotein (LDL) particles supplying the periphery with cholesterol through LDLR-mediated clathrin-dependent endocytosis (for a review, see [2,5]).

LDL catabolism is mediated by hepatic LDLRs, which play a key role in determining plasma LDL cholesterol (LDL-C) levels, as approximately 80% of LDLRs in humans are expressed on hepatocytes [6]. This is exemplified by a case report of a normolipidemic patient who developed severe hypercholesterolemia after receiving a liver transplant from a donor with a heterozygous LDLR mutation [7]. In line with this, children with familial hypercholesterolemia (FH) who underwent liver transplantation had high survival rates and achieved near-normal lipid levels without cardiovascular disease (CVD) progression [8]. In times of cholesterol surplus LDL clearance is suppressed by SREBP2-mediated downregulation of LDLR expression as well as increase in proprotein convertase subtilisin/kexin type 9 (PCSK9) activity, which enhances LDLR degradation [9]. In addition to this, recent findings also support the view that LDLR is sequestered in cholesterol-rich membranes, reducing its affinity for LDL and further impairing LDL clearance under cholesterol-rich conditions [10].

The liver collects excess cholesterol from the periphery either by the above-mentioned receptor-mediated endocytosis of LDL particles [11,12] or via high-density lipoprotein (HDL)-mediated reverse cholesterol transport (RCT) [13]. Additionally, the liver takes up long-circulating oxidized LDL particles [14]. Hepatocytes, the predominant parenchymal cells (PCs) of the liver, play a key role in lipoprotein metabolism. These cells express high levels of LDLR for LDL clearance and scavenger receptor class B type 1 (SR-B1), a high-affinity HDL receptor, which mediates selective cholesteryl ester (CE) uptake [13]. During this process, transient formation of lipid-poor apolipoprotein A1 (ApoA1) occurs as part of its recycling process [15]. In addition, ApoA1 recycling was reported in adipocytes [16]. Excess peripheral cholesterol is transferred to (ApoA1)-rich HDL particles through cholesterol efflux, a process mediated by ATP-binding cassette transporters (ABCA1 and ABCG1). Current findings propose two possible mechanisms for cholesterol efflux: direct transfer at the plasma membrane or endocytosis of ABCA1-ApoA1, followed by lipid collection in endosomes and resecretion through exocytosis [17]. Within HDL, cholesterol is esterified by lecithin–cholesterol acyltransferase (LCAT) and partially exchanged for triglycerides from LDL or VLDL particles via a CETP-mediated process [18,19]. Interestingly, in patients with Tangier disease, a loss of function mutation in ABCA1 which leads to undetectable HDL levels, both LDL and VLDL levels are increased, underling the importance of the CETP-mediated exchange between HDL and LDL for human lipoprotein metabolism [20].

In contrast to vesicular LDL intracellular routes, excess non-esterified lipoprotein-derived cholesterol in the membrane is removed via a non-vesicular pathway [21,22,23,24,25,26]. Free cholesterol derived from HDL particles was proposed to enter cells after desorption by diffusion through the aqueous environment into the cell membrane [27,28,29,30]. The notion that human RCT is different from that in mice comes from tracer studies in humans [31,32,33,34,35,36]. These human studies show that a majority of CE is transported back to the liver by LDL; however, net transport of free cholesterol to the liver by HDL accounted for ~30% of the total plasma-to-liver RCT [31]. Taken together, CE transport in humans is mainly mediated by LDL via CETP, and free cholesterol derived from HDL accounts for a substantial amount of cholesterol delivered to the liver via membrane exchange. In a first step, tethering of lipoprotein particles leads to concentration-dependent cholesterol flux; however, the precise sequence of this process at the plasma membrane is not fully elucidated. Further steps involve sterol binding by the endoplasmic reticulum (ER)-resident proteins Aster-A, Aster-B, and Aster-C, which extract cholesterol from the plasma membrane and transfer it to the ER [37] (for a recent comprehensive review, see [38]). Depending on hepatic cellular cholesterol homeostasis, excess cholesterol can then be converted by Acyl-CoA–cholesterol acyl transferase (ACAT1) into CE for storage in lipid droplets, incorporated into VLDL particles for secretion, or converted into bile acids for excretion. The conversion of excess cholesterol into bile acids represents the final step in RCT [26]. These bile acids are then secreted into the bile ducts, which are lined by cholangiocytes. Both bile acids and cholesterol now pass the enterohepatic circulation [39].

Emerging data highlight the importance of liver sinusoidal endothelial cells (LSECs) in lipoprotein metabolism, alongside contributions from other non-parenchymal cells (NPCs) such as Kupffer cells (KCs) and hepatic stellate cells (HSCs) [40]. LSECs appear to be especially important for removal of oxidized LDL particles [41,42,43]. These LDL particles were recently shown to lead to severe sieve plate loss in LSECs associated with fusion of fenestrae (defenestration) and excessive gap formation [43]. Defenestration and thickening of LSECs associated with increased extracellular matrix deposition are typically observed during aging and are associated with reduced chylomicron remnant clearance, postprandial hyperlipidemia, and reduction in detoxifying function [44,45,46,47]. While defenestration generally is expected to reduce hepatic lipid clearance, it is unclear if the concomitant increased gap formation and collagen matrix accumulation observed by Mao et al. [43] may also contribute by means of non-specific trapping of lipids in the space of Disse to reduced hepatic lipid clearance. Further studies are needed to elucidate the exact impact of these processes on lipoprotein metabolism. In endothelial cells from non-hepatic origins, besides saturable LDL receptor-mediated endocytosis, transcytosis plays an important role, involving activin-like kinase 1 [48] and SR-B1/dedicator of cytokinesis 4 [49,50] (for a review, see [51,52]). In contrast to the anti-atherogenic role of SR-B1 in hepatocytes, where it promotes hepatic lipid clearance as part of RCT, Huang and colleagues [49], for the first-time, have provided evidence of a pro-atherogenic role of SR-B1 in endothelial cells. They demonstrated that SR-B1 mediates active LDL uptake and transcytosis, facilitating LDL accumulation in arterial wall macrophages. These studies further highlight the active role of endothelial cells in lipoprotein metabolism.

Resident liver macrophages, KCs, are a major source of plasma CETP contributing to lipid homeostasis [53]. Strikingly, plasma CETP levels are a predictor of hepatic KC content in humans, which in turn correlate with the extent of liver disease [53]. Second, KCs play a key role in hepatic immune responses modulating liver regeneration as well as lipoprotein uptake in states of metabolic dysfunction-associated fatty liver disease (MAFLD) [54,55,56,57]. In response to, for example, a high-fat diet, KCs release cytokines that act on triglyceride synthesis in hepatocytes, thereby promoting liver steatosis [58]. Another non-parenchymal cell type, HSCs, may also affect lipoprotein metabolism. When quiescent, HSCs store the majority of the body’s vitamin A reserves. However, upon exposure to a high-fat diet, HSCs differentiate into myofibroblasts, producing excessive extracellular matrix deposits and promoting liver fibrosis [59]. The increased risk for both hepatic steatosis and fibrosis, such as in metabolic syndrome disorders, suggests that this excessive matrix deposition, which disrupts normal liver architecture and possibly endothelial fenestration, may, in turn, negatively affect lipoprotein metabolism [51,60].

## 2. Lipoprotein Metabolism in Disease: Mechanistic Insights and Research Limitations in Model Organisms

Alterations in hepatic lipoprotein metabolism play a key role in the development of hypercholesterolemia-associated disorders such as familial hypercholesterolemia (FH) and atherosclerosis, as well as in metabolic syndrome (MetS) disorders [61,62,63], obesity [64], metabolic dysfunction-associated fatty liver disease (MAFLD) [65], diabetes [66], and age-related diseases [67]. Given the central role of hepatic lipoprotein metabolism in these conditions, understanding how specific lipid parameters contribute to disease risk is crucial. In this context, Liu et al. [68] conducted a cohort study with 112,000 subjects, 30% of whom suffered from MAFLD, and found that LDL-C, non-HDL-C, and remnant cholesterol all correlated positively with MAFLD risk, while HDL-C showed a negative correlation. However, the mechanisms underlying these associations remain incompletely understood. Diet-derived oversupply of free fatty acids (FFAs) and glucose, both key drivers of MAFLD, may contribute to these lipid alterations (for a review, see [69]).

The metabolic syndrome is characterized by dyslipoproteinemia, typically presenting as abnormally high levels of triglyceride-rich VLDL particles and reduced levels of HDL. This dysregulation may result from VLDL overproduction, increased HDL catabolism, reduced LDL clearance, or a combination of these factors [61,63]. Not all patients with MetS exhibit insulin resistance, but when it is present, insulin insensitivity enhances lipolysis, leading to increased fatty acid flux to the liver. This in turn, exacerbates hepatic steatosis and VLDL production, thereby worsening metabolic imbalance [66]. Furthermore, hypertriglyceridemia, associated with type 2 diabetes, promotes the formation of large triglyceride (TG)-rich VLDL1 particles, rather than medium-sized VLDL2, and is accompanied by increased levels of highly atherogenic TG remnant particles [70]. These remnants are more prone to conversion into small, dense LDL particles (sdLDLs), a process predominantly mediated by hepatic lipase [5]. LDLs vary in size and density. The atherogenicity of sdLDLs arises from their increased susceptibility to oxidation, higher retention within the arterial wall, and reduced affinity to the LDLR, which prolongs their circulation time [2]. Currently, in addition to absolute LDL-C concentrations, the presence of sdLDLs is considered a major risk factor for atherosclerotic cardiovascular disease (ASCVD), supporting the LDL cumulative exposure hypothesis [71]. Thus, according to contemporary concepts, sdLDLs and TG-rich remnants are recognized as key contributors to ASCVD [5,72].

To provide a deeper understanding of LDL action in disease and to improve targeting therapies, it is essential to elucidate the molecular nature of LDL-LDLR binding. However, for years, this has remained a major challenge due to the size and complexity of LDL particles, the lack of suitable technical methods, and variations in lipoprotein preparations. Recently, a novel approach combining cryo-electron microscopy and artificial intelligence-driven structural modeling has enabled scientists to resolve the structure of ApoB-100, the major component of LDL particles [73]. In this study, Reimund and colleagues [74] demonstrated that LDL binding to LDLR occurs at two distinct interfaces, both involving LDL dimer formation. Furthermore, they have shown that mutations in either ApoB100 or LDLR, both associated with elevated LDL-C levels, are directly mapped to these LDL-LDLR interfaces. This is of particular importance because many of these mutations are linked to FH, a prevalent genetic disorder in humans caused by loss-of-function mutations in LDLR, which leads to an excessive LDL accumulation in the blood stream [75,76]. Homozygous FH patients, and even heterozygous individuals, experience severe elevated LDL-C levels early in life, predisposing them to CVD in midlife and, in severe cases, premature death. FH is an autosomal-dominant genetic disorder associated with mutations in the LDLR, apolipoprotein B (APOB), low-density lipoprotein receptor adapter protein 1 (LDLRAP1), or PCSK9 genes [77]. In a seminal observation in the late 1980s, the group of Scott Grundy showed that the LDL of patients with a heterozygous APOB mutation binds less efficiently to the LDLR (~30% of normal) [78,79]. The defect in APOB resulted in CE-loaded LDL and HDL particles containing the truncated ApoB form [80]. Tracer studies in four FH patients showed that the mean half-life of LDL was dramatically increased from usually 2.9 to 6.6 days [81]. Taken together, the defect in uptake leads to longer circulation of these lipoproteins, and thus the lipoproteins are more prone to oxidation. Indeed, C-reactive protein, a marker for inflammation and oxidative stress, was found to be higher in FH children indicating increased oxidative species at a very early age [82]. LDL particles from FH patients have a higher cholesteryl ester load compared to those of normal subjects [83]. Thus, the LDL of FH patients exhibits a more unfavorable composition, a phenomenon known to accelerate the development of atherosclerosis. In addition to changes in LDL composition, HDL also is altered in FH patients. Schaefer et al. showed by tracing labeled stable isotopes in homozygous FH patients that the detected low levels of HDL and ApoA1 are caused by a combination of increased catabolism and decreased production rate [84]. In heterozygous FH patients the catabolism of HDL was increased, but plasma ApoA1 levels were unaltered due to a compensatory increase in the production rate [85]. In line with this, RCT was reported to be defective in FH patients [86]. Due to the prevalence of FH and limited treatment options, it is important to further elucidate the molecular mechanisms and changes imposed by FH to find more effective intervention strategies. Human hepatic organoids derived from cells of FH patients might be a useful tool.

Lipoprotein (a) [Lp(a)] is another unique lipoprotein secreted by the human liver. The Lp(a) particle is an LDL-like particle that carries ApoB-100 to which an apolipoprotein (a) [apo (a)] is attached via a single disulfide bridge [87]. The liver is the main organ that catabolizes the Lp(a) particle; however, the LDLR and apolipoprotein E (ApoE) have no major role in this process, at least in mice [88,89]. Thus, the mechanism of Lp(a) clearance is still under investigation as both, the LDLR and SR-B1 have been proposed to be involved in Lp(a) clearance [90]. Havekes et al. [91] provided evidence that Lp(a) binds to the LDLR using a fibroblast model. PCSK9 was shown to be involved in this uptake process [92,93]. In line with this, mice overexpressing the LDLR exhibit higher Lp(a) catabolism [94]. However, LDLR knockout mice were reported to have no delayed Lp(a) clearance [88]. Conflicting data have also been reported in humans; some studies have documented higher Lp(a) levels in FH patients [95], while others have reported no difference in Lp(a) levels [96]. These differences might be due to altered expression of scavenger receptors in these patients, as both SR-B1 and cluster of differentiation 36 (CD36) were reported to be involved in its uptake process [97,98]. Lp(a) is the major lipoprotein carrier of oxidized phospholipids, which are proinflammatory and thus increase the atherosclerotic burden. About 85% of lipoprotein-associated oxidized phospholipids circulate along with Lp(a) particles [99]. Taken together, Lp(a) exhibits both proinflammatory and proatherogenic properties and possibly exerts effects on clot stability [100]. Plasma levels of Lp(a) are genetically determined, and high levels are considered an independent risk factor for CVD [101]. Recently it has been shown that on a per-particle basis Lp(a) is approximately six times more atherogenic than LDL itself [102]. As the details of Lp(a) secretion are still not completely understood, human liver organoids, which secrete Lp(a) [103], might be a valid tool to study these processes.

All in all, these findings indicate that impaired LDL uptake leads to prolonged lipoprotein circulation, increasing their susceptibility to oxidation. However, the molecular details regarding how these alterations affect their uptake path await further investigations. Given the prevalence of FH and the limited treatment options, further research into molecular mechanisms underlying FH as well as other metabolic disorders is crucial to developing effective therapeutic strategies.

Thus, experimental model systems that adequately assess human hepatic lipid and lipoprotein-associated pathologies are of significant clinical relevance. In humans, tracer studies and mathematical modeling of hepatic lipoprotein catabolism and secretion have been used to assess changes in lipoprotein metabolism [104]. These studies have been applied to MetS [61,62,63], obesity [64], MAFLD [65], diabetes [105], and FH. According to tracer studies in FH patients, the mean half-life of LDL is dramatically increased [81]. Additionally, low levels of HDL and Apo-AI have been observed in FH patients, resulting from a combination of increased catabolism and decreased production rates [84]. Consequently, RCT was found to be defective in FH patients [86].

Animal models have been instrumental in elucidating lipoprotein receptor pathways, lipid uptake and delivery of lipids to tissues. However, significant species differences exist in lipoprotein metabolism, which must be considered before translating findings to humans (see the overview in Figure 1). For instance, mice lack CETP, resulting in ~80% of their plasma cholesterol being transported in HDL, whereas in humans ~75% of plasma cholesterol is carried in LDL [18]. Thus, murine lipoprotein metabolism differs considerably from that of humans, with mice being naturally resistant to atherosclerosis development [106]. To study the role of the liver in lipoprotein uptake utilizing models with a more human-like lipoprotein metabolism, murine knockout models have been extensively used (for a review, see [107]). One key model is the Ldlr −/− mouse, which exhibits markedly elevated plasma LDL levels and a pronounced increase in LDL cholesterol (LDL-C) in response to dietary cholesterol intake [108]. Similarly, ApoE−/− mice are widely used to study dyslipidemia-associated atherosclerosis in vivo [109,110]. Beyond mice, other animal models such as hamsters, guinea pigs, and rabbits express CETP, resembling a more human-like lipoprotein metabolism [111,112,113]. However, rabbits lack hepatic lipase, which affects their lipid processing [113]. Nonhuman primates closely resemble humans in hepatic lipoprotein metabolism and atherosclerosis development, yet their use is significantly restricted due to ethical concerns [114]. Altogether, these models complement human tracer studies by providing mechanistic insights into lipoprotein metabolism, receptor function, and lipid transport, thereby advancing our understanding of dyslipidemia-driven cardiovascular disease.

## 3. Two-Dimensional In Vitro Cell and Tissue Culture Models for Lipoprotein Metabolism

To overcome the limitations of the above-mentioned models, human hepatic 2D culture models are used to investigate human hepatic lipoprotein metabolism in detail. In this section, we first summarize insights gained from 2D model systems (for an overview, see Figure 1 and Table 1), followed by a focus on the emerging field of complex 3D hepatic organoid models.

### 3.1. Primary Human Hepatocytes and Hepatoma Cell Lines

Most 2D studies have relied on hepatoma cell lines, which exhibit upregulated lipoprotein metabolism due to their high proliferation demands [128,129]. Early investigations elucidating LDL catabolism utilized fibroblasts, including those derived from FH patients, and Chinese Hamster Ovarian Cells [12,115,116,118,122]. In addition, isolated hepatocyte couplets have been effective for electrophysiological studies of bile secretion; however, primary hepatocytes rapidly lose their cellular phenotype [143,144,145]. Bryan Brewer’s group studied LDL metabolism in cultured hepatocytes from normal and homozygous FH subjects and showed that, while normal hepatocytes exhibited saturable LDL binding, FH-derived hepatocytes showed a linear, non-saturable uptake pattern [119,123,179]. Interestingly, their LDL association was reduced to ~30% of normal levels, supporting the existence of alternative LDL uptake pathways. In parallel, Scott Grundy’s group demonstrated that LDL from patients with heterozygous APOB mutations exhibited ~30% reduced LDLR binding efficiency [78,79], with CE-loaded LDL and HDL particles containing a truncated ApoB form [80].

Beyond lipoprotein uptake, the liver secretes ApoB containing lipoproteins in response to dietary cues. Dennis Vance’s group characterized lipoprotein metabolism and secretion in primary human hepatocytes (PHHs) in comparison to primary rodent hepatocytes and hepatoma cell lines [146]. Their studies showed that while cell viability decreased in primary human hepatocytes, they retained their ability to secrete lipoproteins. Their lipoprotein profile secretion was similar to that in human circulation but different in the other two tested cell culture models [146]. Furthermore, hepatoma cell lines were used to characterize ApoE recycling, which initiates ApoE-containing HDL particle formation [130]. In addition, hepatoma cell lines were employed to gain insight into ApoB assembly and secretion [131] and the regulation of the LDLR by PCSK9 [132].

HDL metabolism has been extensively investigated using human hepatoma cell lines that express high levels of SR-B1, which may not fully reflect the situation in the human liver. These cell lines have been beneficial in characterizing selective CE uptake from HDL, where HDL delivers its lipid load without degradation of the particle [120,124,126], as well as in deciphering SR-B1 regulatory pathways [117,133]. Of note, HDL degradation was observed in rat and mouse hepatocytes, which was almost blocked by chloroquine [180,181]. On the other hand, in HepG2 cells, HDL endocytosis has also been described [134,135], a process that can be inhibited by bile acids [136].

Hepatoma cell lines such as HuH-7, HepG2, HuH-6, HepaRG, and Hep3B are cost-effective, easy to culture and expand, highly proliferative, and amenable to genetic modifications. These cells retain to some extent hepatocyte-like polarity to varying degrees [182] and exhibit gene and protein expression patterns similar to PHHs [183]. The choice of hepatoma cell lines should be tailored to the research question. For instance, HepG2 cells develop polarity over time, making them suitable for virus entry studies [182], while HepaRG cells show the highest drug sensitivity, making them ideal for drug-induced hepatotoxicity studies [137]. Despite their utility, these immortalized cell lines have limitations, such as aneuploidy [138], low VLDL secretion [139], and reduced expression of liver-specific genes such as CYP450 enzymes involved in drug metabolism [137,140,141,142]. Further differences between PHHs and hepatoma cell lines include variations in lipidation of ApoB-containing lipoproteins [131,184,185,186,187], glycolytic profiles [188], bile acid synthesis and secretion [189], and urea production [190,191]. In contrast, PHHs are considered the gold standard for studying hepatic functions but have limitations, including a short lifespan in culture, reduced tolerance to experimental perturbations, phenotypic changes over time, and the loss of cell membrane polarity in a 2D environment [[137],[147],[148],,[192]]. Moreover, PHHs are often isolated from diseased donor tissues, which may not represent healthy liver conditions. Due to the mentioned disadvantages or limitations of PHHs and hepatoma cell lines, differentiated hepatocytes from embryonic or induced pluripotent stem cells have been in particular demand in recent years.

### 3.2. Methodological Challenges of iPSC-Derived Hepatocytes

As mentioned above, studies on primary hepatocytes as well as non-parenchymal cells (NPCs), such as liver sinusoidal endothelial cells (LSECs), have been hampered by their de-differentiation and consequent functional loss in culture [40,149]. Thus, in recent years, differentiated parenchymal (hepatocytes) and NPCs derived from induced pluripotent stem cells (iPSCs) or embryonic stem cells (ESCs) have emerged as indispensable tools for studying rare genetic and lipid disorders. They are also widely used in the development of novel lipid-lowering therapies [160,161,162] and in drug screenings for dyslipidemia treatment [152,163,164,165,166].

Human hepatocyte-like cells (HLCs) were first generated more than a decade ago using classical differentiation protocols from iPSCs [153,167,168]. Typically, iPSCs are cultured as 2D monolayers and exposed to a combination of activin, BMP4, and FGF2 or WNT-agonists to induce endoderm formation, followed by hepatic specification. This process often involves hypoxic conditions along with factors such as hepatocyte growth factor (HGF) and the cytokine Oncostatin M [150,153,167,168].

The major challenges of these procedures are the lack of non-parenchymal cell interactions and the absence of a 3D microenvironment. Liver NPCs such as LSECs and HSCs support the expansion of liver progenitor cells that are essential for hepatocyte maturation [154]. Consequently, since the interaction with NPCs is necessary to unleash their full potential, the generated hepatocytes exhibit limited metabolic activity and resemble rather immature hepatocyte-like cells (iHLCs). These fetal liver-like phenotypes are typically characterized by high expression levels of alpha-fetoprotein and lower levels of mature liver-specific metabolic enzymes such as cytochrome P450 [193]. Additionally, contamination with undifferentiated iPSC derivates frequently poses a challenge [168]. Thus, careful characterization of the resulting cell populations is crucial, particularly regarding homogeneity and contamination with undifferentiated iPSC derivates. This is especially important for in vivo transplantation studies to prevent teratoma formation [168].

Researchers have employed different strategies to improve these systems in terms of functionality, such as enriching the liver progenitor cell population using a specific carboxypeptidase peptide marker [194], transducing forkhead box A2 (FOXA2) and hepatocyte nuclear factor 1a (HNF1A) using adenoviral vectors [155], applying alternative differentiation strategies with transforming growth factor beta (TGFβ) receptor antagonists to promote hepatocyte-specific differentiation and suppress undesired lineage differentiation [156], and co-culturing differentiated cells with NPCs [154]. Furthermore, overlaying iPSC-derived HLCs onto a sheet composed of a 3T3 cell layer using three-dimensional micropatterned culture plates [157,158] or culturing them on cellulose nanofibril substrates [159] has been shown to improve hepatocyte functionality, particularly in terms of albumin secretion and CYP activity [157,168]. These findings provide the first evidence that 3D liver organoid systems outperform 2D models.

### 3.3. Applications for iPSC-Derived Hepatocytes

State-of-the-art strategies for generating hepatocytes from iPSCs, as described above, can be utilized to study human hepatocyte functions [153,169,170] and to model liver diseases, particularly in the context of patient-specific conditions (for an overview, see [167]). For example, hiPSC-derived hepatocytes serve as a valid patient-related model to investigate the functions of the LDLR and PCSK9, as well as the underlying disease, FH. This provides a platform for identifying novel therapeutic targets [150,151].

In 2019, Caron et al. differentiated both FH-iPSCs and CRISPR/Cas-mediated genetically corrected FH-iPSCs into hepatocytes. They uncovered that, upon pravastatin treatment, the expression of genes involved in cholesterol metabolism was increased in both cell models and both were permissive to hepatitis C infection, but the virus production in hepatocytes derived from FH-iPSCs was significantly lower comparted to corrected FH-iPSCs-derived hepatocytes, indicating that LDLR has a role in hepatitis C virus morphogenesis [152]. This study highlights how iPSCs, combined with CRISPR/Cas technology, can be used to investigate drug-mediated regulation of genes involved in cholesterol and other metabolic pathways. Other research groups also combined the CRISPR/Cas9 technology with FH patient-derived iPSCs to correct mutations in the LDLR or MTTP gene, aiming to restore the normal lipid levels in iHLCs. In 2017, Cayo et al. used HLCs from a homozygous FH patient to examine 2500 compounds in terms of their lipid-lowering efficacy [166]. Furthermore, Zanoni et al. used iPSC-HLCs derived from homozygous SCARB1-P376L patients to investigate the consequences of the P376L mutation on a cellular level, showing that HDL-C uptake is reduced due to the loss of mature SR-B1 receptors [171]. This demonstrates the value of this methodology, which does not require an invasive liver biopsy for the elucidation of the underlying mechanisms of monogenic disorders. Several research groups have taken the first steps towards an autologous cell-based therapy by demonstrating the feasibility of using episomal plasmids to express physiologically responsive transgenes in genetically deficient iPSCs, thereby restoring LDL uptake [172,173]. Additionally, iHLCs may serve as proof-of-principle models, as exemplified by a study on HSCs derived from FH patients, which showed a lack of LDLR upregulation in response to lovastatin treatment in vitro [151].

Furthermore, in the context of ongoing liver donor shortages for patients with liver injuries and diseases, conditions that place a high socio-economic burden, hiPSC-derived hepatocytes offer a promising solution. They could potentially circumvent liver donor shortage through the transplantation of iHLCs in humans (for a review see, [168]).

### 3.4. Current Approaches for Generating iPSC-Derived Non-Parenchymal Liver Cells

iPSC technology has also been successfully employed to generate liver NPCs. For instance, the generation of LSECs requires a four-stage protocol, including a mesodermal stage; the formation of endothelial progenitors, followed by LSEC progenitors; and finally the application of TGF-ß inhibition under hypoxic conditions to produce LSECs [178]. Similarly, human stellate cells are differentiated from iPSCs using a two-stage protocol, which involves maturation of specific HSC progenitors with high levels of the activated leukocyte cell adhesion molecule (ALCAM) marker [154]. KCs can also be derived from hiPSCs using a 25-day procedure that includes the formation of embryonic bodies as an intermediate step and exposure to hepatic cues [176]. Notably, co-culturing hepatocytes with iKCs has been shown to improve drug sensitivity. This approach, along with platforms developed for the assessment of human tissue-engineered matrices [177], could be utilized to analyze KC physiology in greater depth. In contrast to these complex protocols, a modified, rapid eight-day differentiation protocol for LSECs from iPSCs was recently reported [175], paving the way for simpler and more efficient strategies for liver cell differentiation from iPSCs.

## 4. Human Hepatic 3D In Vitro Models and Other Hepatic Tissue Models

The most common 3D in vitro liver models can be subdivided into two categories: spheroids and organoids (for an overview, see Table 2). While organoids are generally considered as mini-organs in a dish, spheroids typically lack this level of tissue complexity and were originally described as “simple cell aggregates” [195]. However, improved protocols incorporating scaffold structures and hetero-cellular systems have enhanced the complexity and physiological relevance of spheroids. Furthermore, precision-cut liver slices (PCLSs) and liver-on-a-chips are gaining importance as tissue models and will be discussed together with hepatic organoids and spheroids in this section.

### 4.1. Precision-Cut Liver Slices (PCLSs)

Another model that has emerged is the precision-cut liver slice (PCLS) ex vivo culture system. This model uses ex vivo liver explants, preserving intact intercellular and cell-matrix interactions, as well as a well-defined thickness [197]. PCLSs resemble the in vivo pathology more closely than in vitro models, making them ideal for studying hepatic drug metabolism [198,199,200] and liver injuries such as fibrosis [201]. PCLSs are also highly suitable for identifying novel therapeutics (for a review, see [202]). Li et al. used these slices to replicate the initiation of metabolic dysfunction-associated steatotic liver disease (MASLD) to investigate the progression of MASLD and their potential therapeutics [203]. Another group used patient-derived PCLSs to evaluate the therapeutic potential of a novel nuclear factor erythroid-2-related factor 2 (NRF2) activator for the treatment of MAFLD [204]. Others developed a microfluidic-based system that incorporates a microchamber to be able to perform drug metabolism studies with human PCLSs under continuous flow [205]. Ijssennagger et al. generated gene expression profiles in human PCLSs after treatment with Obeticholic acid (OCA), a promising drug for the treatment of MASH and type 2 diabetes that activates the farnesoid X receptor (FXR), a nuclear receptor regulating bile acid, glucose, and cholesterol homeostasis [208]. All these studies showed that PCLSs are an effective model for drug testing and disease research.

One of their main benefits is that the structure and cellular composition of the native liver is preserved. However, a key drawback is the limited access to freshly resected human tissue [202]. Additionally, tissues taken from resected samples are not always entirely healthy, leading to variability in experiments [202]. This issue persists when using slices from diseased tissues, as patients such as those with hepatocellular carcinoma (HCC) may undergo various chemotherapeutic treatments before surgery, leading to liver damage and inconsistent results. Furthermore, the short lifespan of PCLSs, up to 7 days, limits their use in time-intensive experimental approaches. Moreover, hepatic function, such as metabolic activity, can only be maintained for up to 3 days [207]. Another drawback of this system is the repair and regenerative response after slice preparation [206], but PCLSs are nevertheless a valuable tool for the investigation of some liver conditions.

### 4.2. Liver-on-a-Chip Technology

The rapidly evolving liver-on-a-chip technology, which accurately recapitulates key structures and functions of the human liver, is increasingly being used to investigate liver injuries and diseases such as MAFLD by exposing chips to various compounds [209,210,211,212]. For example, Suurmund et al. [210] used this liver-on-a-chip methodology in combination with a steatotic spheroid model composed of hepatocytes and endothelial and Kupffer cells to mimic and monitor the progression of MAFLD. In 2021 another group developed this combined model further by adding stellate cells to achieve a more accurate fibrosis model, and by employing a microfluidic device they were able to gradually administer antifibrotic agents, enabling thereby efficient drug transport and metabolism. Yu et al. [213] demonstrated that combining microfluidic systems with a 3D cell culture technology (spheroids) improved albumin and urea secretion, as well as CYP450 activity, and increased sensitivity to drugs compared to static controls. Furthermore, an imaging study from 2024 investigated the cellular uptake and distribution of Alexa 488-labelled non-targeting and targeting antisense oligonucleotides (ASOs) with or without N-acetylgalactosamine (GalNAc) modification in a liver-on-a-chip model to gain deeper insights into cargo delivery dynamics of therapeutics at a cellular level [214]. Depending on the research question, a variety of liver chips with different microstructures and microfluidic channels have been developed. By seeding multiple cell types into these systems, a 3D environment can be created [215,216]. Additionally, microfluidic channels not only supply cells with nutrients but also expose them to shear forces [217]. The use of microfluidic devices also facilitates the application of therapeutic drugs, and by co-culturing different liver cell types the interactions between these cells during drug exposure can be studied [241]. Organ–organ interactions can even be analyzed by connecting multiple chips. However, despite its potential, microfluidic technology is expensive, time-consuming, and still lacks standardized methods, making it suitable primarily for specific research questions [209].

### 4.3. Three-Dimensional Spheroid Models

Spheroids can be derived from primary cells, such as cryopreserved PHHs from different donors (commercially available sources include BioIVT and KalyCell) [218,219,221], from iPSC-derived HLCs [220,222], or from immortalized hepatic cancer cell lines [223,224,225].

To generate spheroids in scaffold-free conditions, attachment to plates is inhibited by using ultra-low attachment plates, constant rotation, or the hanging-drop method, with or without a supporting medium such as methylcellulose [220]. Kurano et al. [226] showed that HepG2 spheroids prepared with the addition of alginate-beads secreted higher levels of albumin, ApoE, and ApoA-I compared to HepG2 monolayers. By treating the spheroids with a Liver X receptor (LXR) agonist, the production of ApoE-rich HDL particles and VLDL particles was enhanced. Tao et al. used extracellular matrix (ECM) scaffolds, such as Matrigel, to generate ECM-loaded spheroids composed of HepG2 or HuH-7 cell lines, which exhibited higher albumin secretion than the 2D hepatoma cell line control group [227].

Spheroids exhibited improved liver function, with a 10–100-fold increase in CYP450 expression compared to conventional 2D models [220,228]. They have been proposed as a valid model system for studying drug metabolism [219], as well as lipoprotein metabolism in metabolic dysfunction-associated liver diseases, such as hepatic steatosis and insulin resistance [221,229]. Additionally, hepatic stellate cell-derived spheroids have been used to investigate metabolic changes that may promote hepatic fibrosis [230]. For instance, lipid and collagen accumulations in HepG2 and LX-2 (stellate cell) 3D spheroids was reversed by treatment with liraglutide or elafibranor, drugs currently under investigation for MASH in clinal trials [224]. A hetero-cellular spheroid system involving co-cultures of PHHs and primary human liver non-parenchymal cells comprising stellate, endothelial, and CD68-positive cells from different donors was utilized to investigate the pathogenesis of MASH associated with fibrosis and to identify potential drug targets [231]. For modeling MASH-related fibrosis, Holmgren et al. generated a 3D spheroid model by co-culturing human iPSC-derived hepatocytes with human primary stellate cells which can be maintained for up to 35 days and shows increased hepatic functionality compared to 2D co-cultures [222]. However, spheroids have not only been used to model liver conditions, but also to study the mode of action of hepatotoxins or to investigate changes in lipid metabolism caused by a downregulation of lipoprotein homeostasis key players [225,232]. Overall, the 3D spheroid culture environment enhances the functional activity of the PHHs, allowing them to maintain their morphology and functionality for several days, making them ideal tools for liver metabolic function and drug testing.

### 4.4. Liver Organoids: Protocols, Improvements, and Comparisons Between Tissue-Derived and iPSC-Derived Models

Organoids mimic the 3D microenvironment of tissues, closely resembling the functional, structural, and biological complexity of an organ in vitro. Compared to conventional 2D models, they more accurately reflect the physiological state of the tissue [237]. Importantly, organoids enable patient-specific modeling, making them valuable tools for precision medicine, as they are accessible for genetic manipulation and in-depth studies (for a recent review, see [229]).

Generally, organoids are classified into two main categories: those derived from pluripotent stem cells and those originating from tissue-resident stem cells [236]. For the former, human induced pluripotent stem cells (hiPSCs), embryonic stem cells (ESCs), and adult stem cells (ASCs), including mesenchymal stem cells, hepatic progenitor cells, and hematopoietic stem cells, are typically used. In contrast, tissue-derived organoids are generated from stem cells that can be obtained from various model organisms, such as mice, rabbits, guinea pigs, hamsters, and humans (see Figure 2).

In 2015, Huch and colleagues developed a liver organoid model using human liver biopsies from different donors [238]. By seeding cells in 3D scaffolds made from basement membrane extracts and cultivating them in specific expansion media, they achieved an enrichment of cholangiocytes. The authors successfully maintained these liver organoids through serial expansion involving mechanical dissociation for up to six months. Furthermore, the addition of specific hepatocyte differentiation media containing HGF promoted the shift of these liver cultures toward hepatocyte-enriched organoids, a method later employed by others [239]. These liver organoids exhibited glycogen storage capacity, albumin secretion, bile acid production, and ammonia elimination. Importantly, they showed the capacity for LDL uptake, making them suitable for lipoprotein metabolism studies.

Next, Hendriks et al. used fetal liver cells maintained in particular ECM scaffold structures with specific hepatocyte differentiation media to obtain highly proliferative and expandable liver organoids enriched with hepatocytes [240]. These fetal liver-derived organoids were used as a model for liver steatosis in drug target screenings [240]. The authors also established APOB-/- organoids to investigate the influence of genetic factors on MASH, which may therefore serve as invaluable tools for studying different facets of lipoprotein metabolism. However, the major drawbacks of tissue-derived organoids are the limited availability of human samples and the poor resemblance of alternative model organisms to human liver lipid metabolism.

To overcome the above-mentioned drawbacks, the most effective alternative approach for generating hepatic organoids utilizes human pluripotent stem cells (hPSCs), human embryonic stem cells (hESCs), or human induced pluripotent stem cells (hiPSCs). Strategies such as employing Matrigel scaffolds to form endoderm spheres followed by hepatocyte differentiation [196,240], or the simultaneous formation of endoderm and mesoderm with co-differentiation of cholangiocyte-like cells and hepatocyte-like cells to generate hepatobiliary organoids [103,234], are becoming increasingly popular.

Another innovative approach was utilized by Takebe et al. [235], who did not use exogenous scaffolds but instead relied solely on cell–cell interactions and the cell’s inherent ability to self-organize. They mixed hepatic endoderm cells derived from iPSCs with endothelial and mesenchymal cells, allowing them to form liver buds in specialized U-bottom-shaped wells with the support of specific media. Alternatively, this method utilized a mixture of stromal, endothelial, and hepatocyte progenitors derived from iPSCs. These liver buds were fully functional, highly vascularized, and maintained their functionality upon transplantation into mice.

In 2023, Harrison et al. [233] developed a protocol using hPSCs to generate vascularized liver-like organoids in an ECM-independent manner through stepwise addition of small molecules that mimic embryonic liver development (see Figure 1). Hepatic organoid differentiation was initially achieved by forming hPSC aggregates, which were then exposed to a WNT signaling impulse (CHIR99021) to activate the WNT pathway and efficiently differentiate the hPSCs into definitive endoderm over two days. The aggregates were then subjected to hepatoblast differentiation for five days, after which organogenesis was initiated by the interaction of epithelial and mesenchymal populations [242]. This process is driven by multiple paracrine factors, including HGF [243]. Unlike native HGF, the HGF agonist N-hexanoic-Tyr-Ile-6 aminohexanoic amide (dihexa) is chemically stable, making it an ideal substitute for the growth factor to drive hepatocyte differentiation [244]. The protocol described by Harrison et al. [233] is efficient, reproducible, and cost-effective, offering significant advantages over other protocols. The organoids can be generated in a relatively short time (20 days), and the protocol is adaptable to various hESC and hiPSC lines. This liver organoid model was shown to consist of a complex cellular composition, including hepatocytes, cholangiocytes, endothelial cells, stellate cells, and Kupffer cells. It also formed bile ducts, biliary networks, and vascular structures (see Figure 2 for a graphical illustration).

Overall, liver organoids exhibit key hepatic functions, such as albumin secretion, bilirubin uptake, glycogen storage, urea synthesis, CYP450 activity, LDL uptake, and the internalization and secretion of ApoB [103,196,233,234,235]. Since the expression patterns of these liver genes vary across different organoid generation protocols, the choice of method should be tailored to the specific research questions.

### 4.5. Exploring Lipoprotein Metabolism: Potential and Applicability of Liver Organoids

Liver-like organoid model systems are perfectly suited for analyzing human hepatic lipoprotein metabolism. Indeed, Harrison et al. [233] have shown that the expression of several apolipoproteins (ApoA1, ApoA4, ApoC3, and ApoD), the presence of scavenger receptors (LYVE, LRP1, CD36, and SCARB1), and the uptake of acetylated LDL by LSEC populations make liver organoids excellent tools for studying lipoprotein metabolism in vitro. For example, fluorescently labeled lipoprotein particles [134] can be added to liver organoids to investigate uptake with spatial and temporal route resolution, as well as cell-type specificity (see Figure 2).

The applicability of this approach has been demonstrated by Roudaut et al. [103], who showed that liver organoids were capable of taking up fluorescently labeled LDL particles as well as lipids in general. In addition to the expression of apolipoprotein B, they also observed the expression of apolipoprotein (a), which is present in highly atherogenic Lp(a) particles. Importantly, the authors found that secreted Apo(a) levels were 50-fold higher in 3D liver organoids compared to 2D models derived from patients with genetically elevated plasma concentrations of Lp(a), highlighting the potential of these for lipoprotein uptake studies [103]. Consistent with this, it was recently shown that liver organoids generated on hyaluronic acid-based hydroscaffolds can internalize and secrete lipoproteins, including Lp(a) [89]. This opens the possibility of studying the contribution of different liver cell types to lipoprotein uptake and its regulation in great detail by using such liver organoid models. Furthermore, the sequential steps and key checkpoints involved in the uptake of various lipoprotein particles, including modified forms such as Lp(a) and oxidized LDL particles, can be systematically analyzed. For example, a recently identified macropinocytosis pathway mediating Lp(a) endocytosis in HepG2 cells, which involves multiple plasminogen receptors, could be validated in physiologically relevant liver organoid systems [89].

Moreover, with the use of advanced techniques such as single-particle profiling, the biophysical properties of lipoproteins can be examined, potentially providing insights into lipoprotein changes in liver organoids derived from different donors [245,246]. Furthermore, liver organoids can be used to analyze the interaction of HDL and LDL in their uptake processes. Particularly, in this context the hepatic WASH complex that was identified as a regulator of LDL and HDL metabolism in hepatocytes may be analyzed [247] (see Figure 2). Another example is given in view of the microRNA (miR)-mediated regulatory mechanism controlling LDLR abundance, as recently shown in human hepatocytes [248]. Here, liver organoids may offer a perfect system to study how miRs affect hepatic lipoprotein uptake. Additionally, alterations in hepatic lipoprotein metabolism induced by genetic diseases like FH can be studied in detail. To achieve specific gene inactivation, the CRISPR/Cas9 techniques could also be applied in liver organoid models. With this tool, essential genes/receptors like LRP1 could be targeted in order to examine their role in lipoprotein uptake. Receptors like the LDLR or SR-B1 are known to facilitate entry of viruses such as Hepatitis C Virus (HCV) to the liver (for a review, see [249]). Thus, by using liver organoids the interplay of different cell types as well as their distinct lipoprotein uptake mechanisms may be analyzed in detail.

The mode of action and signaling pathways affected in this process could be determined using hepatic organoids if the latter are incubated in the presence of high amounts of FFAs and glucose to drive lipid droplet formation. Additionally, those substances could be used in hepatic organoid systems to drive the transformation in the direction of MASH and to study the progression towards hepatocellular carcinoma (HCC) by controlled activation of oncogenes in addition to pharmacological or genetic means. In this regard, the role of endoplasmic reticulum stress is known to be particularly important for the progression of MAFLD [250] and could be analyzed in more depth. Furthermore, important insights could be gained into mechanisms of how MAFLD progresses to steatohepatitis and cirrhosis. Finally, this hepatic model system can be used in different disease settings such as liver cancer. Thereby different spatial–temporal stages of cancer may be investigated using hiPSCs harboring a set of selected oncogenes. Presumably, liver organoids covering diseases such as FH and MAFLD [251,252,253] will be available in the near future, highlighting the vast possibilities of such liver organoids.

Liver organoids also play a crucial role in regenerative medicine and provide a platform for high-throughput gene manipulation studies [240]. iPSC-derived as well as tissue-derived liver organoids [240] have also been successfully used in oncology, particularly for studying Hepatitis B Virus (HBV)-associated carcinogenesis [239]. Notably, liver organoids are becoming commercially available (e.g., via Molecular Devices or Stemcell), though human liver organoids are not readily available on the market. Currently, all commercially available liver organoids are either derived from mice or cancer patients. As a proof of concept, we recently adapted the protocol of Harrison et al. [233] to generate human liver organoids. Using hiPSCs derived from patients with Hutchinson–Gilford progeria syndrome (HGPS), a premature aging disease, we generated liver organoids that could be used to study age-related changes in lipoprotein metabolism or even liver aging in vitro. This exemplifies how the availability of iPSCs, which can be used to generate liver organoids, enables rapid and efficient studies of lipoprotein metabolism in a dish. In the future, this approach could be extended to the currently available iPSC collection, comprising 171 different iPSC lines modeling different lipoprotein-based diseases, deposited by the Rader lab. These include iPSCs derived from individuals with FH, Tangier disease, abetalipoproteinemia, and hyperalphalipoproteinemia and individuals with no reported diagnoses (www.wicell.org, accessed on 20 June 2025).

## 5. Future Directions

Organoids are emerging as valuable model systems for studying metabolic processes. Despite significant advancements, many aspects of human hepatic metabolism and its regulatory mechanisms remain poorly understood. Hepatic organoids provide a powerful platform to investigate the interplay between different liver cell types and their roles in liver metabolism.

Liver organoids offer not just the perfect opportunity to study normal liver physiology and alterations in hepatic metabolism in a dish, but also facilitate research into complex pathological processes, such as those associated with aging, cancer, and liver diseases induced by viruses or hypercaloric diets. In addition to representing a physiologically relevant human-like system, the liver organoid approach reduces the burden of heavily used animal research, which is highly desirable and aligns with ethical principles of replacement, reduction, and refinement (3R). The applicability of liver organoids is further expanded by their adaptability to different genetic backgrounds. As demonstrated in this review, the use of HGPS patient-derived liver organoids exemplifies their potential as powerful tools in precision medicine. Despite their advantages, liver organoids still have limitations. While some models are vascularized, they do not fully replicate the complete nervous and vascular systems of the human liver. In addition, organoid cultures are more time-consuming to generate than spheroids. However, ongoing advancements are continuously improving these systems. Innovations such as microfluidic systems help mimic blood flow and create concentration gradients for nutrients and oxygen, enabling the study of spatial differences in liver lipoprotein metabolism. However, as the area of stem cell research and organoid technology development is rapidly evolving, one can envision that these challenges will be progressively addressed in the future.

In summary, human liver organoids represent a powerful tool to analyze metabolic processes, as well as the crosstalk between different liver cell types during lipoprotein uptake in the human liver in situ. Alongside liver organoids, other model systems such as liver spheroids, various hepatoma cell lines, and iPSC-derived 2D hepatocyte cultures modeling different lipoprotein diseases also provide valuable insights into lipoprotein metabolism. These systems enable the study of complex signaling pathways regulating lipoprotein homeostasis, allowing for genetic manipulations in a dish to decipher key regulatory mechanisms.

## Figures and Tables

**Figure 1 biomolecules-15-00956-f001:**
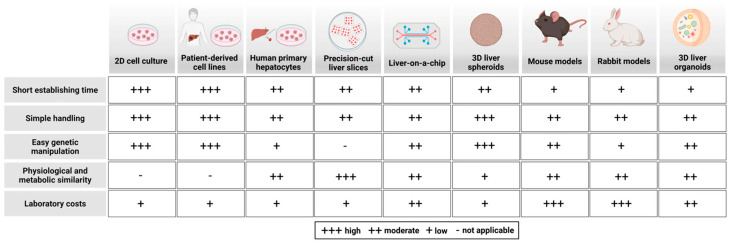
Comparison of model systems with regard to hepatic lipoprotein metabolism.

**Figure 2 biomolecules-15-00956-f002:**
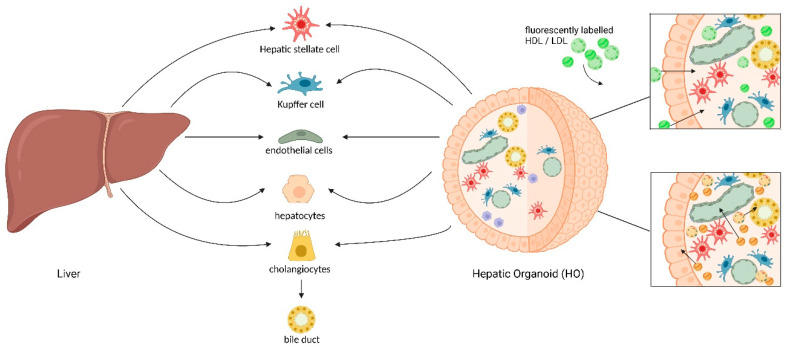
Cell-type composition of the human liver and of pluripotent stem cell-derived liver organoids generated by the current state-of-the-art technology. The human liver consists of several cell types, including hepatocytes, biliary epithelial cells/cholangiocytes forming the bile ducts, stellate cells, Kupffer cells, and liver sinusoidal endothelial cells. Each of these cell types possesses unique functions that cooperatively regulate the hepatic function. Hepatocytes make up the majority of the liver volume and perform many of the functions ascribed to the hepatic metabolism. Pluripotent stem cell-derived liver organoids contain all the major cell types of liver tissue, as demonstrated by several groups. These may be used to study lipoprotein metabolism extrinsically by adding fluorescently labeled lipoprotein particles (upper right) or intrinsically by analysis of endogenously produced lipoproteins (lower right). All figures were created with BioRender.com.

**Table 1 biomolecules-15-00956-t001:** Two-dimensional in vitro cell and tissue culture models used to study hepatic lipoprotein metabolism and the corresponding diseases.

Source	Achievements/Use	Advantages	Drawbacks	References
CHO cells	Analysis of cholesterol metabolism and LDL catabolism	Easy to handle	Not a valid model for liver metabolism	[86,115,116,117]
Fibroblasts/patient-derived fibroblasts	Detailed LDL uptake studies in normal and FH-derived fibroblasts	Reflect disease, easy to cultivate	Do not reflect all facets of hepatic metabolism	[12,79,118,119,120,121,122,123,124]
Hepatic cell lines/hepatoma cell lines (HepG2, HUH6, HUH7, HeparRG, Hep-3B)	Detailed studies on lipoprotein metabolism, e.g., ApoE recycling, ApoB assembly and secretion, LDLR-PCSK9 axis, SR-B1 function in selective cholesterol ester uptake, differences between LDL and HDL uptake	Non-limited proliferation, easy genetic manipulations, partial hepatocyte polarization, to some extent similar gene expression profile, can be used for a liver-on-a-chip model	Non-physiological, cancer origin, aneuploidy, low VLDL secretion, unphysiologically high SR-B1 levels, altered ApoB lipidation, low CYP450, preferential anaerobic glycolysis, altered bile acid and urea synthesis	[124,125,126,127,128,129,130,131,132,133,134,135,136,137,138,139,140,141,142]
Primary hepatocytes/patient-derived hepatocytes	Linear non-saturable LDL uptake (FH patients), alternative uptake routes (FH patients)	Fresh primary cultures, reflect lipoprotein disease and physiological models, can be used for a liver-on-a-chip model	Limited human donors, diseased donor tissue poorly reflects physiological lipid metabolism, loss of polarization upon culture, less efficient genetic manipulations	[117,119,123,143,144,145,146,147,148]
Hepatic non-parenchymal cells	Delineation of their role in the maintenance of hepatic homeostasis	Analysis of their role in lipoprotein metabolism, detailed analysis possible in combination with a chip and by co-culturing with PC cells	Polarization generally lost after isolation, lack of interaction with hepatocytes and lack of 3D microenvironment	[40,41,42,43,53,54,55,56,57,58,59,149]
iPSC-derived hepatocytes	Proof-of-principle studies, e.g., FH disease model, deciphering PCSK9 mutations and function	Circumvent liver donor shortage problem, very good liver disease models, generation directly from patients possible, recapitulate to high extent human hepatocyte metabolism, genetic manipulations possible, can be co-cultured with NPCs, can used for liver-on-a-chip	Varying degrees of hepatocyte-like cell phenotype (dependent on the method of generation), absence of 3D microenvironment, possible contamination with undifferentiated iPSC derivates	[150,151,152,153,154,155,156,157,158,159,160,161,162,163,164,165,166,167,168,169,170,171,172,173]
iPSC-derived non-parenchymal cells	Establishment of functional human liver model in vitro from LPCs, LSECs, and HSCs derived from hiPSCs	Analysis of cell-specific functions like vitamin A storage, CETP production, or fibrosis development possible; can be used in combination with other cell types and a chip to create a 3D microenvironment	Absent 3D microenvironment, possible contamination with undifferentiated iPSC derivates	[154,174,175,176,177,178]

**Table 2 biomolecules-15-00956-t002:** Three-dimensional model systems used to study hepatic lipoprotein metabolism and the corresponding diseases.

Source	Achievements/Use	Advantages	Drawbacks	References
Precision-cut liver slices	Used to study liver injuries and hepatic drug metabolism, suitable for the identification of novel therapeutics	Intact intercellular and cell-matrix interactions, resembles the in vivo pathology, preserves structural and cellular composition of native liver	Limited access to freshly resected human tissue, slices are usually from diseased tissues, short lifespan, repair and regenerative response after slice preparation	[196,197,198,199,200,201,202,203,204,205,206,207,208]
Liver-on-a-chip	Valid model for the analysis of hepatic lipoprotein metabolism, used to investigate liver injuries and diseases, suitable for drug screenings	Recapitulates key structure and function of native liver, a variety of chips with different microstructures and-channels are available, a 3D environment can be created, easy administration of nutrients and drugs due to a microfluidic system	Very expensive, time-consuming, lacks applicable standard methods	[209,210,211,212,213,214,215,216,217]
Spheroids (PHHs; HLCs; immortalized cancer cell lines, e.g., HepG2 and NPCs)	Good models to study drug metabolism and metabolic changes	Improved liver functions due to 3D environment, easy generation, co-culturing possible	Ethical constraints, limited number of donors, genetic variability, cancer origin, not all liver cell types included	[195,218,219,220,221,222,223,224,225,226,227,228,229,230,231,232]
PSC-derived liver organoids (hESCs, hiPSCs)	Good models to study hepatic metabolism and liver diseases	Consist of more or all liver cell types, exhibit key liver functions (e.g., albumin secretion) and structure (e.g., bile ducts and vascular network), accessible to genetic manipulation, can be used for regenerative medicine, efficient, reproducible, no genetic variability	Long generation time (e.g., 20 days)	[103,233,234,235]
Tissue-derived hepatic organoids	Good models to study liver diseases and drug metabolism	Reflect disease, consist of (all) liver cell types	Ethical constraints, usually diseased tissues, limited human sample availability, genetic variability	[236,237,238,239,240]

## Data Availability

No new data were created or analyzed in this study. Data sharing is not applicable to this article.

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
