# Peer review of "Hepatic Lipoprotein Metabolism: Current and Future In Vitro Cell-Based Systems"

_biomolecules, 2025, doi:10.3390/biom15070956_

Round 1
Reviewer 1 Report
Comments and Suggestions for Authors
Manuscript ID: Biomolecules-3671152
Title: Hepatic lipoprotein metabolism: state-of-the-art concepts and emerging
study tools
Authors: Izabella Kiss, Nicole Neuwert, Raimund Oberle, Markus Hengstschläger, Selma Osmanagic-Myers *, and Herbert Stangl *
This is a thorough review article that provides a detailed description of hepatic lipoprotein metabolism in healthy and disease states followed by an examination of available cell-based model systems used to interrogate liver lipoprotein biology/metabolism. A comprehensive and informative review is presented, however some outstanding issues remain to be addressed prior to publication in Biomolecules.
- The title of this review article is somewhat vague. This review focuses on current and future in vitro cell based systems to investigate hepatic lipoprotein metabolism. This should be conveyed in the title.
- While the review is comprehensive in content, it comes across as ‘dense’ for the reader. The inclusion of a few conceptual figures and schemes would make the review more ‘palatable’ for the reader.
- Table 1 is very small and also illegible. The authors should make this table larger.
- Tables 2 and 3 present various 2D and 3D systems followed by discussions of the systems respectively. However the order of the discussion of cell systems are not in sequence as presented in the table (particularly for table 3). This should be corrected where the order of systems presented in the table should be discussed in the same order.
- Inclusion of data/figure from selected referenced studies would also strengthen the review.
Author Response
Reviewer 1
This is a thorough review article that provides a detailed description of hepatic lipoprotein metabolism in healthy and disease states followed by an examination of available cell-based model systems used to interrogate liver lipoprotein biology/metabolism. A comprehensive and informative review is presented, however some outstanding issues remain to be addressed prior to publication in Biomolecules.
Answer: We thank the reviewer for the kind evaluation.
- The title of this review article is somewhat vague. This review focuses on current and future in vitro cell based systems to investigate hepatic lipoprotein metabolism. This should be conveyed in the title.
Answer: We have altered the title according to the suggestion of the reviewer.
- While the review is comprehensive in content, it comes across as ‘dense’ for the reader. The inclusion of a few conceptual figures and schemes would make the review more ‘palatable’ for the reader. Table 1 is very small and also illegible. The authors should make this table larger. Tables 2 and 3 present various 2D and 3D systems followed by discussions of the systems respectively. However the order of the discussion of cell systems are not in sequence as presented in the table (particularly for table 3). This should be corrected where the order of systems presented in the table should be discussed in the same order. Inclusion of data/figure from selected referenced studies would also strengthen the review.
Answer: We have increased the size of Table 1, however, due to input requirements in the template supplied by MDPI we cannot increase its size further. We have changed the order in Table 3 according to the flow of the manuscript. To guide the reader, we presented a Graphical Abstract, which should attract readers to the review and also guide them through the main text, from 2D approaches to more complex organoids. In addition, Table 1 and Figure 1 aim to guide the reader.

Reviewer 2 Report
Comments and Suggestions for Authors
This is a well written paper that covers most of the titled topic is sufficient detail and appropriate references. There were some minor deficiencies as follows:
Line 61: Cite add Liver transplantation to provide LDLR that lowered plasma cholesterol in a child with homozygous FH.
Line 76: Add a likely role for recycling of transiently formed lipid-free APOA1.
Introduction should also address the role of spontaneous HDL-cholesterol transfer according to PMID: 7104306
Also, a large fraction of cholesterol is hepatically removed in the unesterified form halftimes of less than ten minutes. See PMID: 15145983. Free cholesterol transfer to macrophages is a function of HDL-free cholesterol content. PMID: 39566848
292: Note degradation of HDL by hepatocytes in vitro.
Some syntax issues as follows:
Line 63 and other places: When possible, use active rather than passive voice for better impact. E.g., “In times of cholesterol surplus there is a suppression of LDL clearance through SREBP2-mediated à In times of cholesterol surplus LDL clearance is suppressed by SREBP2-mediated …”
Line 67: sequestered in cholesterol-rich membranes, reducing its affinity bioavailability for to LDL and further impairing LDL clearance under cholesterol-rich conditions ….
Line 122: crophages, KCs, were shown to be are the
Line 123: were even demonstrated to be a predictor predict …
Line 130: these cells serve as are the major storage site for vitamin A
Line 227: According to tracer studies in FH patients have demonstrated that the mean
The word demonstrate is overused. Revealed, uncovered, discovered instead.
Author Response
Reviewer 2
This is a well written paper that covers most of the titled topic is sufficient detail and appropriate references. There were some minor deficiencies as follows:
Answer: We thank the reviewer for the kind evaluation. We have changed the manuscript according to the reviewers’ suggestions (see below).
Line 61: Cite add Liver transplantation to provide LDLR that lowered plasma cholesterol in a child with homozygous FH.
Answer: We added a line on the therapeutic effect of liver transplantation in children.
Line 76: Add a likely role for recycling of transiently formed lipid-free APOA1.
Answer: We added a sentence on apo-A1 recycling in introduction (marked yellow).
Introduction should also address the role of spontaneous HDL-cholesterol transfer according to PMID: 7104306
Also, a large fraction of cholesterol is hepatically removed in the unesterified form halftimes of less than ten minutes. See PMID: 15145983. Free cholesterol transfer to macrophages is a function of HDL-free cholesterol content. PMID: 39566848
Answer: We extended the paragraph on transfer of free cholesterol via aqueous diffusion and added several references including the one’s mentioned by the reviewer (lines 111 -120).
292: Note degradation of HDL by hepatocytes in vitro.
Answer: We agree with the reviewer that there are conflicting data on HDL degradation; uptake of HDL containing ApoE will be mediated by the LDLR family, however there are some reports on degradation of HDL by liver hepatocytes. We have added a sentence on HDL degradation in hepatocytes (line 386-388).
Some syntax issues as follows:
Line 63 and other places: When possible, use active rather than passive voice for better impact. E.g., “In times of cholesterol surplus there is a suppression of LDL clearance through SREBP2-mediated à In times of cholesterol surplus LDL clearance is suppressed by SREBP2-mediated …”
Line 67: sequestered in cholesterol-rich membranes, reducing its affinity bioavailability for to LDL and further impairing LDL clearance under cholesterol-rich conditions ….
Line 122: crophages, KCs, were shown to be are the
Line 123: were even demonstrated to be a predictor predict …
Line 130: these cells serve as are the major storage site for vitamin A
Line 227: According to tracer studies in FH patients have demonstrated that the mean
The word demonstrate is overused. Revealed, uncovered, discovered instead.
Answer: We have corrected the errors accordingly.

Reviewer 3 Report
Comments and Suggestions for Authors
Kiss et al. summarize in their review the role of the liver in lipoprotein metabolism with major emphasis on state of the art methods in in vitro models . The manuscript highlights the significance of liver in humans and in experimental animals.
COMMENTS
This is very comprehensive review of general interest. The manuscript is concisely and well understandable presented and considers the recent literature on this topic.
There is one point that that needs attention by the authors:
Lp(a), a lipoprotein that is currently in the focus of lipoprotein metabolism is almost exclusively expressed in the liver. There are hardly any animal models that express Lp(a) and thus in vitro models with liver cells appear to be ideal instruments in that respect. Apolipoprotein (a) is mentioned in this review only very superficially and the authors are advised to elaborate on this topic more intensely. At minimum, one ¶ summarizing the significance of the liver in Lp(a) metabolism should be inserted in the Introduction for readers who are not so familiar with this lipoprotein.
Author Response
Reviewer 3
Kiss et al. summarize in their review the role of the liver in lipoprotein metabolism with major emphasis on state of the art methods in in vitro models . The manuscript highlights the significance of liver in humans and in experimental animals.
This is very comprehensive review of general interest. The manuscript is concisely and well understandable presented and considers the recent literature on this topic
Answer: We thank the reviewer for the kind evaluation.
There is one point that that needs attention by the authors:
Lp(a), a lipoprotein that is currently in the focus of lipoprotein metabolism is almost exclusively expressed in the liver. There are hardly any animal models that express Lp(a) and thus in vitro models with liver cells appear to be ideal instruments in that respect. Apolipoprotein (a) is mentioned in this review only very superficially and the authors are advised to elaborate on this topic more intensely. At minimum, one ¶ summarizing the significance of the liver in Lp(a) metabolism should be inserted in the Introduction for readers who are not so familiar with this lipoprotein.
Answer: We have extended the paragraph on Lp(a) (lines 266- 293). As the reviewer kindly pointed out, the importance o the liver organoid model for Lp(a) research was already discussed in the original manuscript (lines 764- 783).